# Diminishing neuronal acidification by channelrhodopsins with low proton conduction

Rebecca Frank Hayward[1], F Phil Brooks III[2], Shang Yang[3], Shiqiang Gao[3], Adam E Cohen[2,4]*

[1]School of Engineering and Applied Sciences, Harvard University, Cambridge, United States; [2]Department of Chemistry, Harvard University, Cambridge, United States; [3]Department of Neurophysiology, University of Wurzburg, Wurzburg, Germany; [4]Department of Physics, Harvard University, Cambridge, United States

**\*For correspondence:**
cohen@chemistry.harvard.edu

**Abstract** Many channelrhodopsins are permeable to protons. We found that in neurons, activation of a high-current channelrhodopsin, CheRiff, led to significant acidification, with faster acidification in the dendrites than in the soma. Experiments with patterned optogenetic stimulation in monolayers of HEK cells established that the acidification was due to proton transport through the opsin, rather than through other voltage-dependent channels. We identified and characterized two opsins which showed large photocurrents, but small proton permeability, PsCatCh2.0 and ChR2-3M. PsCatCh2.0 showed excellent response kinetics and was also spectrally compatible with simultaneous voltage imaging with QuasAr6a. Stimulation-evoked acidification is a possible source of disruptions to cell health in scientific and prospective therapeutic applications of optogenetics. Channelrhodopsins with low proton permeability are a promising strategy for avoiding these problems.

## eLife assessment

This **important** and **compelling** study investigates the problem of intracellular acidification induced by commonly-used optogenetic stimulating opsins. The low proton permeability of two high-performance opsins is shown to reduce photostimulated acidification. The findings may be of broad interest in the fields of neuroscience research and optogenetic therapies.

## Introduction

Channelrhodopsins are light-gated ion channels that are widely used to modulate the activity of neurons and other excitable cells (*Govorunova et al., 2021*). In addition to research use, these tools are entering clinical practice as a treatment for forms blindness (*Busskamp et al., 2012*; *Sahel et al., 2021*) and are under consideration for treatments of other disorders of neural excitability (*Chow and Boyden, 2013*; *Krook-Magnuson et al., 2013*; *Bansal et al., 2023*). Every ion channel carries current through one or more ions, and so the induced change in membrane voltage is always accompanied by a change in ionic concentrations. In both research and clinical applications, one must consider whether the ionic changes in the cell have effects beyond the purely electrical effects of the channel.

Ionic perturbations are largest when (1) the basal intracellular concentration of the relevant ion is low, (2) the surface-to-volume ratio is high, and (3) the channel is activated chronically. Sodium, potassium, and chloride ion concentrations in cells are typically in the millimolar range, and thus the fractional changes in concentration of these ions due to channel opening are typically <1%. In contrast,

protons and calcium ions have low free concentrations, typically ~100 nM or lower. In these cases, the ionic fluxes due to channel opening can substantially perturb the concentration.

The amount of charge flow required to change membrane voltage by a given amount is proportional to the capacitance, and hence the membrane surface area. This charge is diluted into the volume. For this reason, ionic concentrations in small structures with high surface-to-volume ratios are more labile to optogenetic perturbations than are concentrations in large structures. One must therefore consider whether optogenetic tools substantially perturb ionic concentrations in thin structures such as axons, dendrites, or dendritic spines. Finally, ionic concentrations equilibrate much more slowly than does membrane potential. Brief optogenetic stimuli may have negligible effects on concentrations, whereas chronic stimuli with high duty cycle may have cumulative effects. Concerns about long-term consequences of optogenetic stimulation are particularly relevant to prospective therapeutic applications, where the tools may be used over long times in patients.

Channelrhodopsins have been shown to acidify cells, while light-driven outward proton pumps, such as Archaerhodopsin 3, alkalize cells (*Beppu et al., 2014*; *Bo et al., 2020*). Indeed, optogenetically triggered alkalization has been proposed as a tool to control cell death (*Nakao et al., 2022*). Intracellular acidification can suppress neuronal excitability and also suppress vesicle release (*Sinning et al., 2011*; *Xiang and Bergold, 2000*; *Sinning and Hübner, 2013*) but has also been reported to enhance release of adenosine (*Lee et al., 1996*; *Dulla et al., 2009*) and dopamine (*Cannizzaro et al., 2003*). Changes in intracellular pH can also affect cell differentiation (*Oginuma et al., 2020*) and metabolism (*Fang et al., 2017*) and survival (*Beppu et al., 2014*; *Bo et al., 2020*; *Xiang and Bergold, 2000*; *Lam et al., 2013*). Nominally similar channelrhodopsins have been reported to evoke opposite behavioral effects in live mice (*Baleisyte et al., 2022*). The cause of these differences is not known, but it is possible that these differences could be due, in part, to differences in ionic selectivity.

For these reasons, it is important to quantify and ultimately minimize perturbations to cellular pH from optogenetic tools. We combined channelrhodopsin stimulation with a red-shifted fluorescent pH sensor, pHoran4 (*Shen et al., 2014*), for measurement of pH changes during optogenetic stimulation. We used patterned optogenetic stimulation in gap junction-coupled cellular monolayers to establish that the acidification was due to proton flux directly through the channelrhodopsin. We then tested two new opsins, ChR2-3M and PsCatCh2.0 (*Chen et al., 2022*), with a low proton permeability and found minimal perturbations to cellular pH. We performed a detailed electrophysiological and photophysical characterization of these opsins and showed that they are compatible with simultaneous voltage imaging. The new opsins may be promising for clinical applications where acidification is undesirable.

## Results
### CheRiff acidifies neurons

We developed lentiviral constructs and optical stimulus protocols for simultaneous optogenetic stimulation and pH measurements (*Figure 1A*). For the actuator we used CheRiff-GFP, a non-selective cation channel with an activation peak at 460 nm (*Hochbaum et al., 2014*). For the pH measurement we used pHoran4 (*Shen et al., 2014*), a red-shifted reporter with an excitation peak at 547 nm and a $pK_a$ of 7.5. We calibrated the pH response of pHoran4 in permeabilized HEK293T (HEK) cells and in cultured neurons (Methods; *Figure 1—figure supplement 1*) and then used this calibration to convert changes in fluorescence to changes in pH. We assumed an initial pH of 7.3 for all the cells (*Figure 1B*). Since our measurements focused on relative pH changes as opposed to absolute pH, modest deviations from this assumption would not affect the interpretation of the following results.

We imaged pH changes in cultured neurons during CheRiff stimulation. We alternated epochs of optogenetic stimulation (488 nm, 400–800 mW cm$^{-2}$, 0.5 s) and pH imaging (561 nm, 100–200 mW cm$^{-2}$, 1 s) to avoid crosstalk of the blue light into the pH recordings (*Figure 1C*). The pH dynamics were much slower than 1.5 s, so this process did not sacrifice information.

After 150 s of stimulation and imaging, the pH had decreased within the neurons that expressed CheRiff-GFP, from 7.3 to 6.76 ± 0.35 (mean ± standard deviation [SD], n = 34 cells, *Figure 1D, E*), corresponding to an approximately threefold increase in concentration of free protons. We then measured the pH recovery for an additional 150 s without optogenetic stimulation (*Figure 1E*). Recovery was slow, returning to only pH 6.95 ± 0.33 (mean ± SD) after 150 s. Control experiments in neurons

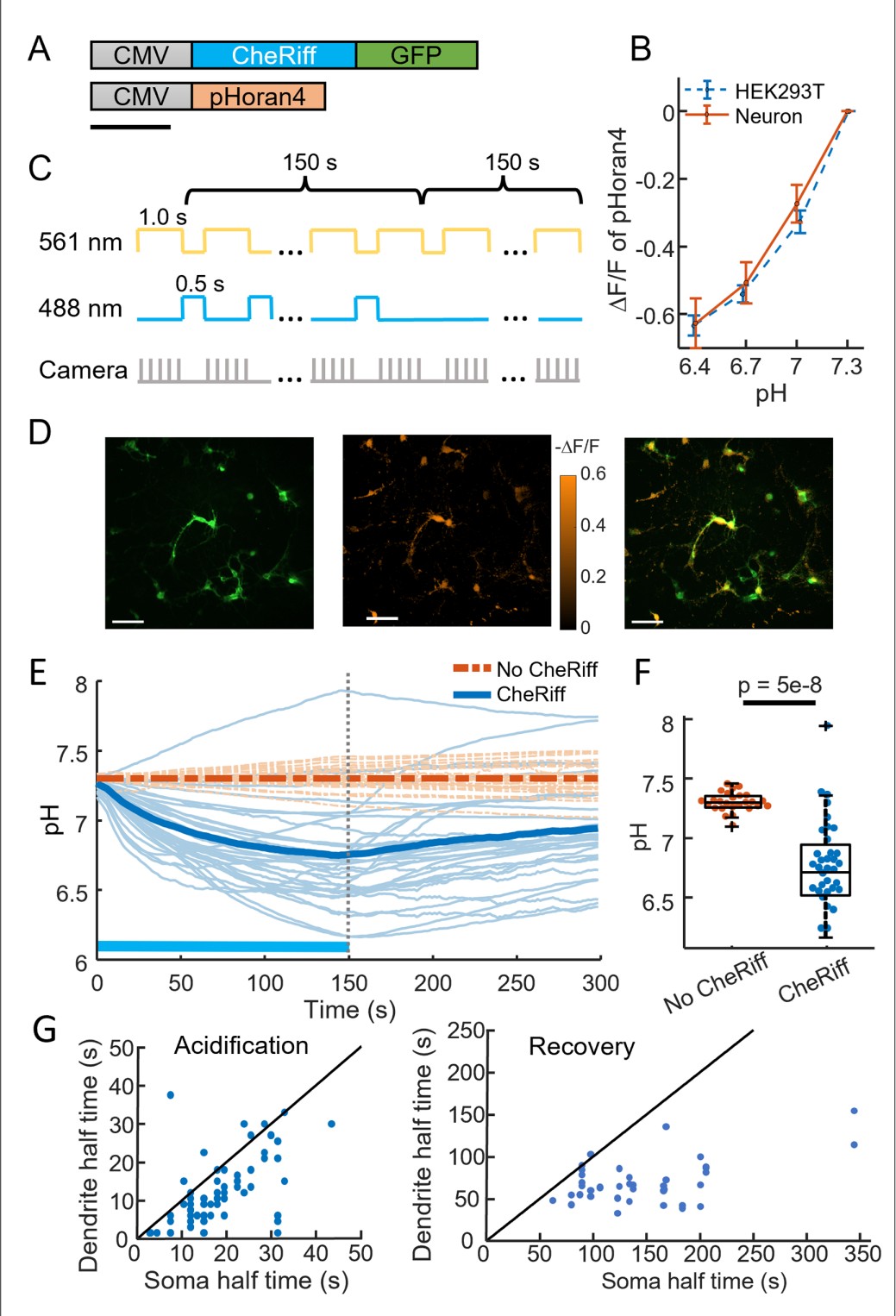

**Figure 1.** CheRiff acidifies polarized cells. (**A**) Genetic constructs for simultaneous optogenetic stimulation and pH imaging. (**B**) Calibration of pHoran4 pH sensor in HEK cells and neurons. Error bars represent standard deviation (SD) of $n = 8$ measurements in HEK cells, 42 measurements in neurons. (**C**) Protocol for measuring pH responses to optogenetic stimulation. Stimulation (blue) and measurement (yellow) were interleaved for 150 s; then pH recovery was measured for 150 s without optogenetic stimulation. (**D**) Example images of cultured neurons showing (left) Green Fluorescent Protein (GFP) fluorescence, a marker for CheRiff expression, (middle) $-\Delta F/F$ in the pHoran4

*Figure 1 continued on next page*

*Figure 1 continued*

channel after 150 s of the protocol shown in (**C**), (right) merge. Scale bars 100 μm. (**E**) Time-course of pH in cultured neurons. Cells expressing pHoran4 but not CheRiff did not acidify. Bold lines show population average. (**F**) CheRiff-expressing neurons acidified to a pH of 6.76 ± 0.35 (mean ± SD, *n* = 34 cells). Neurons not expressing CheRiff had significantly less acidification, pH = 7.3 ± 0.08 (mean ± SD, *n* = 26 cells, p = 5e−8 Wilcoxon rank sum test). Box plots show inter-quartile ranges, tick-marks show data range, + shows outlier. (**G**) Half-time of (left) acidification or (right) recovery for neuron somas vs dendrites stimulated with the protocol in (**C**). Black line shows equal kinetics.

The online version of this article includes the following figure supplement(s) for figure 1:

**Figure supplement 1.** Procedure for calibrating pHoran4 pH measurements.

that expressed pHoran4 but not CheRiff showed a small fluorescence increase during the stimulation period (0.11 Δ*F/F*), much smaller in magnitude and opposite in sign compared to the change in cells expressing CheRiff and pHoran4 (−0.34 Δ*F/F*). We attribute the slight increase in fluorescence of the CheRiff-negative cells to a blue light-mediated photoactivation artifact, as has been seen in other fluorescent reporters (*Farhi et al., 2019*). The mean photoartifact from CheRiff-negative cells was subtracted from all recordings of CheRiff-positive cells prior to analysis.

We observed that stimulation-induced pH changes and post-stimulation recovery were faster in the dendrites than in the soma (*Figure 1G*). This effect is most likely due to the higher surface-to-volume ratio of thin processes. For a given proton current density across the membrane, the change in local proton concentration is greater in a thin tube than in the large soma.

## Acidification is via proton transport through the opsin

We next sought to determine to what extent the acidification was due to proton transport through the opsin vs through depolarization-induced opening of endogenous proton-permeable channels or other activity-dependent acidification mechanisms (e.g. metabolic shifts). Working in HEK cells, we expressed CheRiff, pHoran4, and a doxycycline-inducible inward rectifying potassium channel, $K_{ir}2.1$ (Methods, *Figure 2A*). The $K_{ir}2.1$ channel polarized the HEK cells to a resting potential of approximately −70 mV (*Zhang et al., 2016*), providing a driving force for proton entry. $K_{ir}2.1$ has been reported not to carry a proton current itself (*Ye et al., 2016*). When the cells were grown into a confluent monolayer, they coupled electrically via endogenous gap junctions (*Figure 2B*; *McNamara et al., 2016*).

We characterized these cultures using a wide-area 'Firefly' microscope which provided spatially and temporally patterned illumination at 470 and 561 nm via a digital micromirror device (DMD) (*Werley et al., 2017*). *Figure 2C* shows the protocol for interleaved CheRiff stimulation and pH imaging. After 60 s of stimulation, cells expressing all three components were acidified to a pH of 6.96 ± 0.15 (mean ± SD, *n* = 75 cells). Cells not expressing $K_{ir}2.1$ had substantially less acidification (final pH 7.21 ± 0.05, mean ± SD, *n* = 70 cells, p = 2e−24, Wilcoxon signed-rank test), confirming the importance of membrane voltage as a driving force for proton entry. Cells lacking both the CheRiff and the $K_{ir}2.1$ did not show detectable acidification (final pH 7.31 ± 0.02, mean ± SD, *n* = 13 cells, *Figure 2D–F*), consistent with our results in neurons (*Figure 1E, F*).

These results established that acidification occurred in non-excitable cells but left open the possibility that the HEK cells might contain an endogenous proton conductance that opened upon membrane depolarization. To test this possibility, we took advantage of the gap junctional coupling between cells in a confluent monolayer. Due to the gap junctions, local CheRiff activation led to depolarization of neighboring regions, with an electrotonic length constant of ~300 μm (*McNamara et al., 2016*). While protons can also diffuse through gap junctions, this process is orders of magnitude slower than propagation of membrane voltage (*Wu et al., 2019*). This difference in lateral electrical vs proton transport permitted us to indirectly depolarize cells via gap junction coupling, and to ask whether the acidification arose in all depolarized cells or only in cells with direct CheRiff activation.

We used the DMD to pattern the optogenetic stimulation into stripes, with a separation between the stripes of 95 μm, much smaller than the electrotonic length constant (*Figure 2G*). We used the far-red voltage sensitive dye BeRST1 (*Huang et al., 2015*) to map the voltage changes throughout a monolayer of HEK cells expressing pHoran4, CheRiff, and $K_{ir}2.1$. *Figure 2H* shows the stimulus pattern (reported via fluorescence of CheRiff-GFP), and the electrical depolarization pattern. As expected

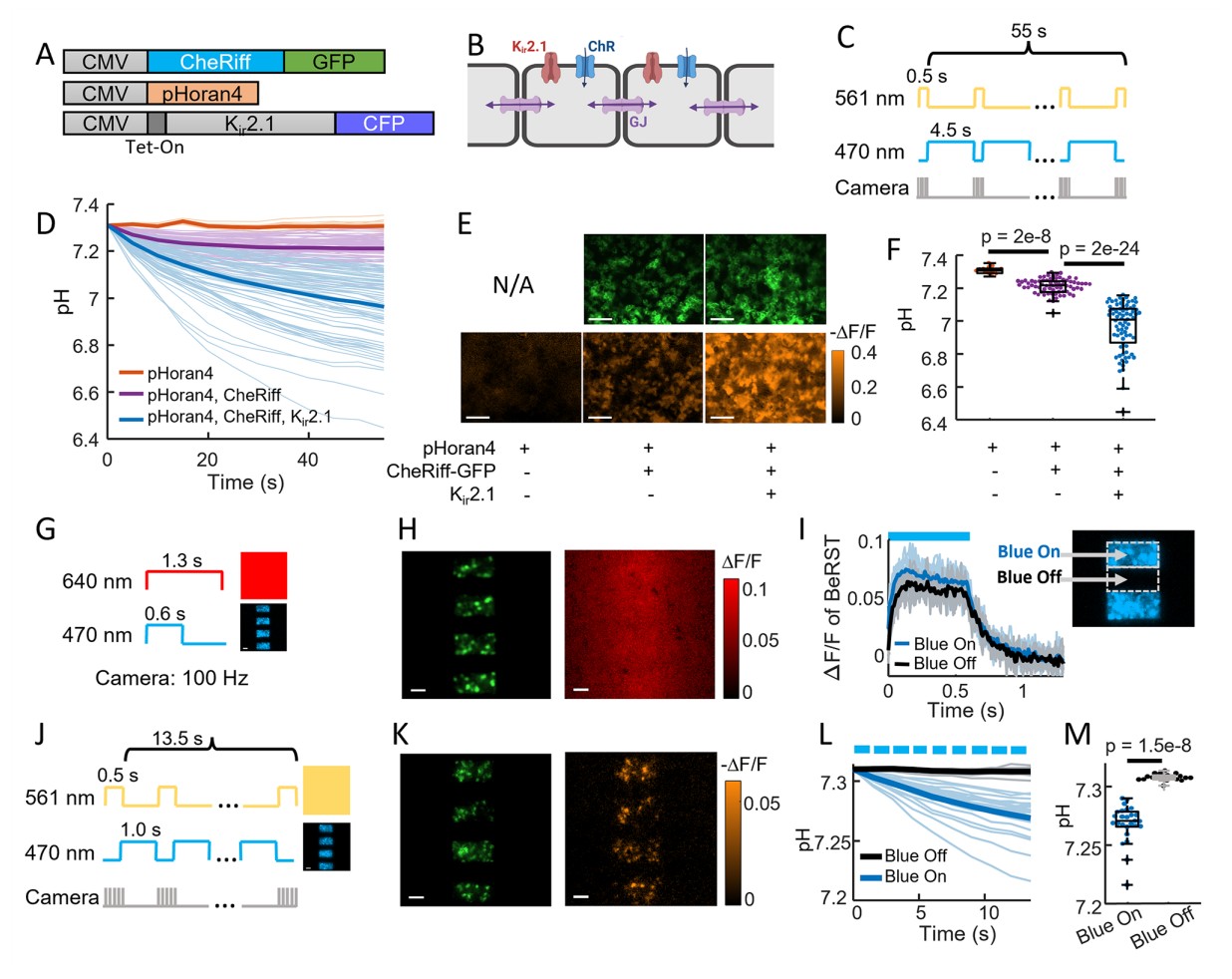

**Figure 2.** CheRiff exhibits high proton conductance. (**A**) Genetic constructs for simultaneous optogenetic stimulation and pH imaging in polarized HEK293T cells. (**B**) Diagram of HEK cell monolayer connected by gap junctions. (**C**) Experimental paradigm for measuring pH responses to optogenetic stimulation. Stimulation (488 nm) and measurement (561 nm) were interleaved to avoid optical crosstalk. (**D**) Time-course of pH in HEK cells. Expression of Kir2.1 increased the driving force for proton influx, substantially enhancing the acidification. (**E**) Images of HEK cell monolayers showing (top) GFP fluorescence, a marker for CheRiff expression and (bottom) $-\Delta F/F$ in the pHoran4 channel after protocol shoin (**C**). Scale bars 100 μm. (**F**) Quantification of the data in (**D–E**). pHoran4 alone: pH = 7.31 ± 0.02 (mean ± standard deviation [SD], $n$ = 13 cells); CheRiff and pHoran4 pH = 7.21 ± 0.05 ($n$ = 70 cells); CheRiff, pHoran4, and Kir2.1: pH = 6.96 ± 0.15 ($n$ = 75 cells). Statistical comparisons via Wilcoxon signed-rank test. (**G**) Protocol for mapping voltage responses to patterned optogenetic stimulation (488 nm) via fluorescence of BeRST1 (640 nm exc.). (**H**) Images of HEK cell monolayers showing (left) fluorescence of GFP with patterned blue illumination, (right) $\Delta F/F$ of BeRST1. Scale bars 100 μm. (**I**) Time-course of BeRST1 fluorescence in HEK cells inside (Blue On) and outside (Blue Off) the optogenetic stimulus regions. (**J**) Protocol for measuring pH responses to patterned optogenetic stimulation. Stimulation (488 nm) and measurement (561 nm) were interleaved to avoid optical crosstalk. (**K**) (Left) Fluorescence of GFP with patterned blue illumination, (right) $\Delta F/F$ in the pHoran4 channel after protocol shown in (**J**). Scale bars 100 μm. (**L**) Time-course of pH inside (Blue On) and outside (Blue Off) the optogenetic stimulus regions. (**M**) Quantification of the data in (**L**). Directly stimulated cells acidified to pH = 7.27 ± 0.016 (mean ± SD, $n$ = 26 cells), indirectly depolarized cells (Blue Off) did not acidify: pH = 7.31 ± 0.003 ($n$ = 19 cells, $p$ = 1.5e−8 Wilcoxon signed-rank test).

from the strong gap junctional coupling, the optogenetically induced depolarization in the interstitial 'Blue Off' regions was almost as large as in the directly stimulated 'Blue On' regions (**Figure 2I**).

We then mapped the pH changes over the whole field of view, using the same striped stimulation pattern alternating with wide-field yellow illumination for pH imaging (**Figure 2J**). We expected that proton currents through CheRiff would follow the illumination pattern precisely, whereas proton currents through voltage-gated channels would follow the much smoother voltage profile. We quantified acidification after only a short (13.5 s) period of CheRiff stimulation to avoid possible confounding effects of lateral proton diffusion between cells. In the directly stimulated 'Blue On' regions we observed robust acidification after 13.5 s (pH = 7.27 ± 0.016, mean ± SD, $n$ = 26 cells), and in the

indirectly depolarized 'Blue Off' regions we observed no acidification (pH = 7.31 ± 0.003, mean ± SD, $n$ = 19 cells, p = 1.5e−8 Wilcoxon signed-rank test, *Figure 2K–M*). These results establish that CheRiff directly acidifies polarized cells via proton transport through the opsin, and that electrical depolarization alone is insufficient to drive acidification.

## ChR2-3M and PsCatCh2.0 are potent non-acidifying channelrhodopsins

Several channelrhodopsins were recently reported to have low proton conductivity (*Chen et al., 2022*; *Duan et al., 2019*; *Vierock et al., 2017*; *Hososhima et al., 2020*; *Scholz et al., 2017*), so we tested two of these for acidification in HEK cells and characterized their photophysical properties. PsCatCh2.0 (*Chen et al., 2022*) is derived from the highly blue-shifted *Platymonas subcordiformis* PsChR (*Govorunova et al., 2013*) via the L115C mutation and addition of a trafficking signal, ER export signal, and cleavable N-terminal Lucy-Rho signal peptide. Due to its high speed and high light sensitivity, this opsin has been used for visual function restoration in blind mice (*Chen et al., 2022*).

The second opsin we tested is derived from the recently engineered ChR2-XXM (i.e. ChR2-D156H), which shows high photocurrent and high selectivity for $Na^+$ and $K^+$ over $H^+$ (*Duan et al., 2019*; *Scholz et al., 2017*). Mutating H134 to Q at the intracellular gate further enhanced the $Na^+$ and $K^+$ selectivity (*Figure 3—figure supplement 1*) and photocurrent amplitude. Mutating E101 to N near the extracellular gate site also boosted the $Na^+$ and $K^+$ selectivity (*Figure 3—figure supplement 1A*) without affecting the photocurrent amplitude (*Figure 3—figure supplement 1B*). To optimize expression and trafficking, we added the same trafficking, ER export, and signal peptides as in PsCatCh2.0. We designate this triple mutant of ChR2 as 'ChR2-3M'.

Following the same procedure as in *Figure 2C*, we tested the acidification due to opsin stimulation in electrically polarized HEK cells expressing pHoran4, $K_{ir}2.1$, and either CheRiff, ChR2-3M, or PsCatCh2.0 (*Figure 3A*). After a 55-s stimulation and imaging protocol, CheRiff cells showed a decrease in pH as above (pH = 6.98 ± 0.15, mean ± SD, $n$ = 170 cells), while the new opsins did not show any significant changes in pH (ChR2-3M: pH = 7.31 ± 0.10, $n$ = 63 cells; PsCatCh2.0: pH = 7.30 ± 0.03, $n$ = 74 cells; p = 4e−31, p = 4e−35, Wilcoxon signed-rank test; *Figure 3B–D*). These experiments confirmed the low proton permeability of ChR2-3M and PsCatCh2.0. In paired experiments, we used voltage imaging with BeRST1 to confirm that all three opsins induced depolarization in the HEK cell monolayers (*Figure 3—figure supplement 2*).

We then performed a detailed characterization of the new opsins using patch clamp electrophysiology in HEK cells (*Figure 3E–I*, *Table 1*). Both opsins were sensitive to blue (488 nm) light. ChR2-3M passed unusually large steady-state photocurrents (1378 ± 618 pA, mean ± SD, $n$ = 4 cells) and was highly sensitive to blue light (effective power density for 50% activation, EPD50 = 11.6 ± 8.7 mW cm$^{-2}$, *Figure 3E*), but had slow opening ($\tau_{on}$ = 57 ± 21 ms at saturating blue light) and very slow closing ($\tau_{off}$ = 1950 ± 500 ms, *Figure 3F*). PsCatCh2.0 had somewhat smaller photocurrents (847 ± 359 pA, $n$ = 6 cells), higher EPD50 (116 ± 13 mW cm$^{-2}$), but very fast opening ($\tau_{on}$ = 4.2 ± 3.5 ms at saturating blue light) and closing ($\tau_{off}$ = 17.6 ± 3.4 ms).

Both opsins had positive reversal potentials (ChR2-3M: 16.6 ± 3.6 mV, PsCatCh2.0: 12.3 ± 4.7 mV), consistent with preferential $Na^+$ and $Ca^{2+}$ selectivity (*Figure 3G*). A key attraction of PsCatCh2.0 for quantitative optogenetics experiments was its very flat-top response to a step in blue light. In contrast to CheRiff which shows substantial sag in photocurrent upon continuous illumination (*Hochbaum et al., 2014*), PsCatCh2.0 showed almost no sag. This low extent of light-induced inactivation appears to be, at least in part, a characteristic of this particular type of opsin from *P. subcordiformis* (*Govorunova et al., 2013*).

For applications in all-optical electrophysiology (i.e. simultaneous stimulation and voltage or calcium imaging), it is critical that the light used for imaging a red-shifted reporter does not interfere with the action of the opsin. At 50 mW cm$^{-2}$ excitation intensity, ChR2-3M retained substantial activation at 561 nm (20%) and 594 nm (4.8%), but undetectable activation at 640 nm (<0.3%, *Figure 3H*). PsCatCh2.0 was more promising for all-optical applications: at 561 nm the photocurrent was only 2% and at 594 and 640 nm the photocurrent was undetectable (<0.3%).

In some opsins, light at a red-shifted wavelength can reverse retinal isomerization, forcing the channel closed (*Berndt et al., 2009*; *Venkatachalam et al., 2014*). To mimic the conditions of a typical all-optical electrophysiology experiment, we thus tested the combination of blue (240 mW cm$^{-2}$) and intense orange (594 nm, 1 W cm$^{-2}$) or red (640 nm, 8 W cm$^{-2}$) light (*Figure 3I*). The orange light had

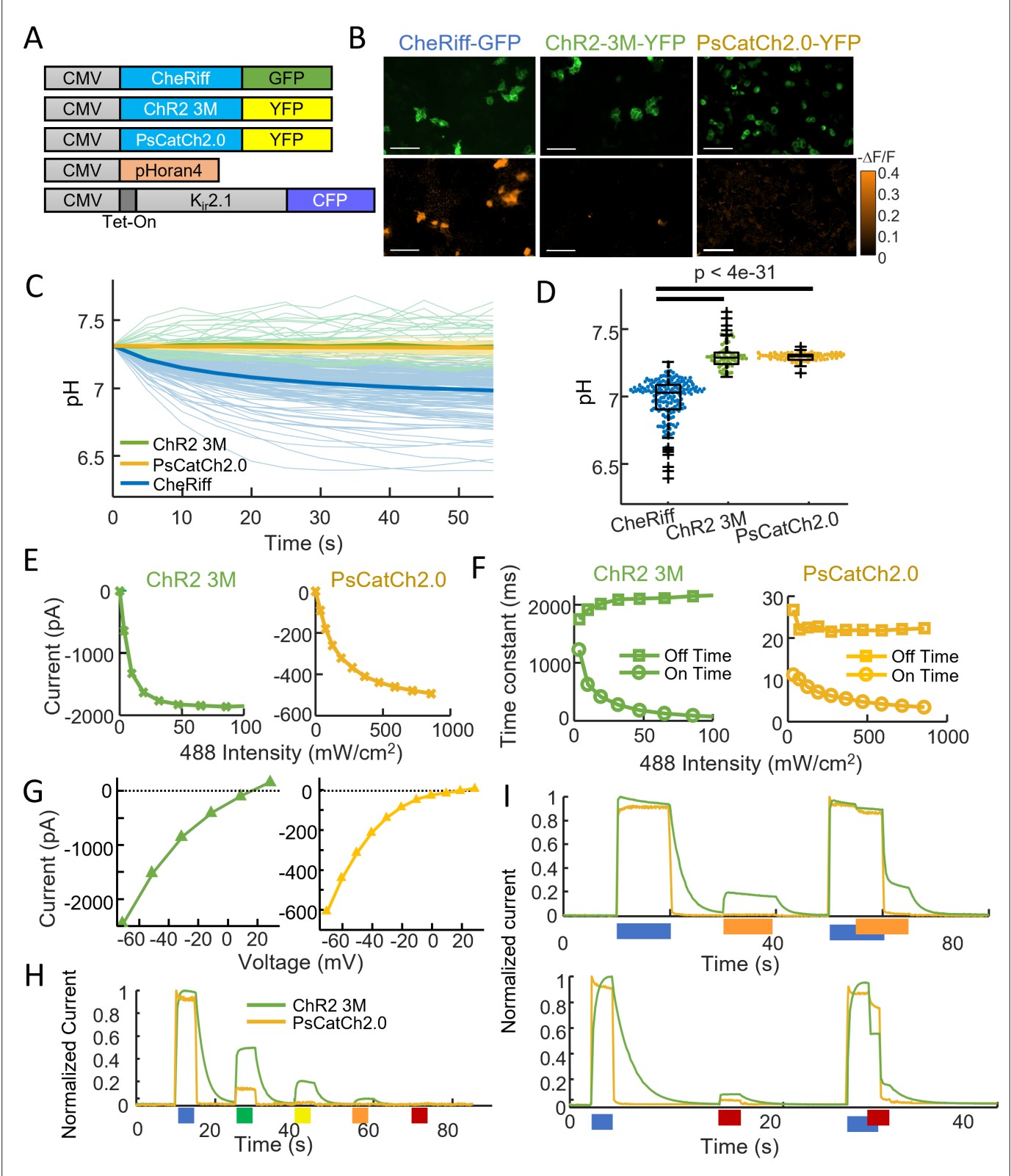

**Figure 3.** ChR2-3M and PsCatCh2.0 are potent non-acidifying channelrhodopsins. (**A**) Genetic constructs for simultaneous optogenetic stimulation using channelrhodopsin variants and pH imaging in polarized HEK cells. (**B**) Images of HEK cells showing (top) GFP or YFP fluorescence, a marker for channelrhodopsin expression and (bottom) −Δ*F/F* in the pHoran4 channel, measured after protocol shown in **Figure 2C**. Scale bars 100 μm. (**C**) Time-course of pH in HEK cells expressing the three opsins. (**D**) Quantification of theta in (**C**). CheRiff: pH = 6.98 ± 0.15 (mean ± standard deviation [SD], *n* =

*Figure 3 continued on next page*

*Figure 3 continued*

170 cells); ChR2-3M: pH = 7.31 ± 0.10 (*n* = 63 cells); PsCatCh2.0: pH = 7.30 ± 0.03 (*n* = 74 cells); p = 4e−31, p = 4e−35, Wilcoxon signed-rank test. (**E–I**) Whole-cell voltage clamp measurements on HEK cells expressing channelrhodopsins. (**E**) Steady-state photocurrents as a function of blue illumination intensity. (**F**) Opening and closing kinetics as a function of blue light intensity. (**G**) Steady-state photocurrents as a function of holding voltage. (**H**) Normalized photocurrents from stimulation with light at 488, 532, 561, 594, and 640 nm (50 mW cm$^{-2}$ in all cases). (**I**) Normalized photocurrents from combinations of blue (488 nm, 240 mW cm$^{-2}$) and orange (594 nm, 1 W cm$^{-2}$) or red (640 nm, 8 W cm$^{-2}$) light corresponding to intensities typical for all-optical electrophysiology.

The online version of this article includes the following figure supplement(s) for figure 3:

**Figure supplement 1.** Engineering of ChR2-3M, a channelrhodopsin with high Na$^+$ and K$^+$ selectivity and high photocurrent amplitude.

**Figure supplement 2.** Depolarization of HEK cell monolayers via patterned stimulation of channelrhodopsins.

negligible activating or inactivating crosstalk into PsCatCh2.0 activation, but partially activated the ChR2-3M (~20%). The red light slightly activated both constructs (~10% for ChR2-3M and ~5% for PsCatCh2.0), and also substantially inactivated ChR2-3M, leading to a ~40% drop in photocurrent. Together, these results indicate that PsCatCh2.0 is a particularly promising channelrhodopsin for all-optical physiology experiments.

We then tested the new opsins in cultured rat hippocampal neurons (*Figure 4*). Under paired optogenetic stimulation and pH imaging (*Figure 4A, B*), we observed acidification in cells expressing CheRiff (pH = 6.87 ± 0.27, mean ± SD, *n* = 24 cells), as before. We observed substantially less acidification in cells expressing either ChR2-3M (pH = 7.13 ± 0.19, mean ± SD, *n* = 31 cells, p = 2.5e−4), or PsCatCh (pH = 7.14 ± 0.11, mean ± SD, *n* = 25 cells, p = 4e−5, *Figure 4C–E*). We then produced cultures co-expressing each of the three channelrhodopsins and QuasAr6a for voltage imaging (*Figure 4F, G*). Under blue light stimulation, each opsin-induced reliable spiking (*Figure 4H*). ChR2-3M also induced some firing in intervals after a blue light pulse, presumably due to the very slow closing of the channel ($\tau_{off}$ = 1950 ± 500 ms, *Figure 3F*) leading to residual currents.

## Discussion

Most polarized cells have a strong inward-directed proton-motive force, and sodium-proton exchangers are required to maintain physiological intracellular pH (*Wakabayashi et al., 1997*; *Hoffmann and Simonsen, 1989*). A sudden change in proton permeability of the membrane can disrupt this balance, leading to intracellular acidification. The proton flux was sufficient to overwhelm the buffering capacity of the cytoplasm. While we focused on the acidification due to CheRiff, we expect similar levels of acidification from other channelrhodopsins, unless they have been engineered specifically to be proton-impermeable.

The degree of acidification depends on the subcellular opsin distribution, the cell geometry, and the stimulus protocol. In general, the smaller the compartment, the larger the acidification. While we could not resolve individual dendritic spines, our results suggest that if channerhodopsins exist in spines, optogenetic acidification in spines could be substantial. In some all-optical physiology experiments, channelrhodopsins are restricted to the soma and proximal dendrites via trafficking sequences such as the K$_v$2.1 trafficking motif (*Adam et al., 2019*). While this restriction has been primarily to facilitate targeted single-cell activation, a possible side effect is to decrease acidification in small distal compartments (dendrites, spines, and axon).

**Table 1.** Comparison of channelrhodopsin gating properties.

EPD50 is the effective power density for 50% activation. CheRiff data are from Fig. S9 and Table S4 of *Hochbaum et al., 2014*. CheRiff reversal potential is from *Zhang et al., 2016*.

| | Reversal potential (mV) | $t_{on}$ fastest (ms) | $t_{on}$ at EPD50 (ms) | $t_{off}$ (ms) | EPD50 (mW cm$^{-2}$) | Steady-state photocurrent (pA) | $I_{peak}/I_{SS}$ |
|---|---|---|---|---|---|---|---|
| CheRiff | 4 | 4.5 ± 0.3 | - | 16 ± 0.8 | 22 ± 4 | 1300 ± 80 | 0.65 |
| ChR2-3M (*n* = 4) | 16.6 ± 3.6 | 57 ± 21 | 800 ± 550 | 1950 ± 500 | 11.6 ± 8.7 | 1378 ± 618 | 1.00 |
| PsCatCh2.0 (*n* = 6) | 12.3 ± 4.7 | 4.2 ± 3.5 | 9.3 ± 1.2 | 17.6 ± 3.4 | 116 ±1 3 | 847 ± 359 | 0.92 |

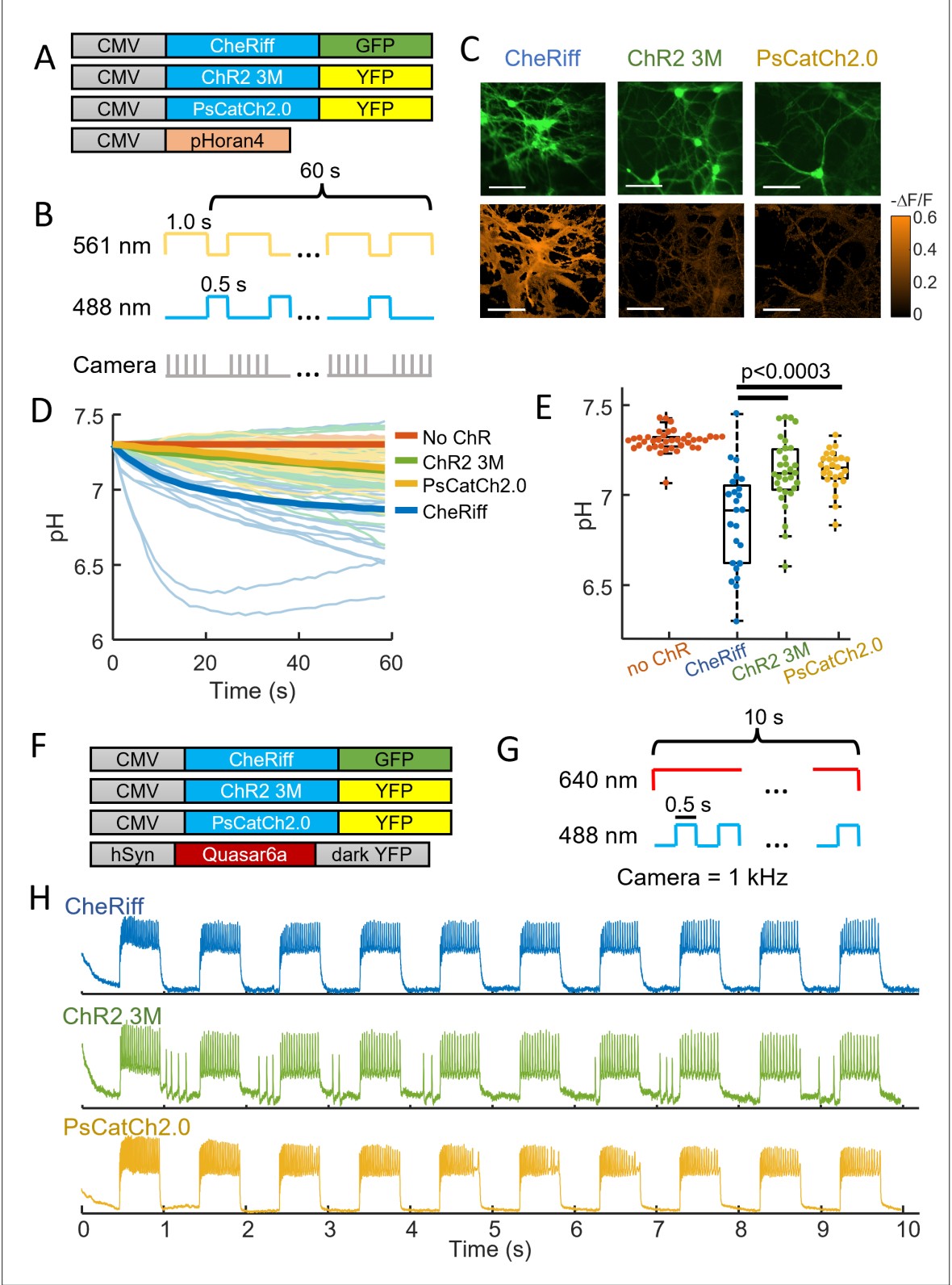

**Figure 4.** Chr2-3M and PsCatCh2.0 acidify neurons less than CheRiff. (**A**) Genetic constructs for simultaneous optogenetic stimulation and pH imaging. (**B**) Experimental paradigm for measuring pH responses to optogenetic stimulation. Stimulation (blue) and measurement (yellow) are interleaved for 60 s to avoid optical crosstalk. (**C**) Images of cultured neurons showing (top) GFP or YFP fluorescence, a marker for channelrhodopsin expression, (bottom) $\Delta F/F$ in the pHoran4 channel after the protocol shown in (**B**). (left) CheRiff-GFP, (middle) ChR2-3M-YFP, (right) PsCatCh2.0. Scale bars 100 μm. (**D**) Time-

*Figure 4 continued on next page*

*Figure 4 continued*

course of pH dynamics in cultured neurons. Cells expressing ChR2-3M and PsCatCh2.0 acidify less than CheRiff. (**E**) Neurons expressing ChR2-3M, pH = 7.13 ± 0.19 (mean ± standard deviation [SD], n = 31 cells), and PsCatCh2.0, pH = 7.14 ± 0.11 (mean ± SD, n = 25 cells) had significantly less acidification (p = 2.5e−4, p = 4e−5), respectively, (Wilcoxon signed-rank test) than CheRiff-expressing neurons, pH of 6.87 ± 0.27 (mean ± SD, n = 24 cells). (**F**) Genetic constructs for simultaneous optogenetic stimulation and voltage imaging. (**G**) Experimental paradigm for measuring voltage responses to optogenetic stimulation. Stimulation (blue) and measurement (red). (**H**) Time-course of optogenetically activated spiking in cultured neuron expressing (top) CheRiff, (middle) ChR2-3M, or (bottom) PsCatCh2.0.

Here, we showed that the high-performance opsins ChR2-3M and PsCatCh2.0 have very low proton permeability, enabling repeated stimulation with minimal local acidification. We observed no activation-induced acidification in HEK cells (*Figure 3C, D*), but we did observe a very slight acidification in neurons (*Figure 4D, E*). We speculate that the activity-induced neuronal acidification was due to cell autonomous mechanism (*Chesler and Kaila, 1992*; *Chesler, 2003*; *Svichar et al., 2011*) as opposed to proton transport through the opsins.

While the ChR2-3M construct is not optimal for all-optical physiology experiments due to crosstalk with longer wavelength light, several features make it promising for prospective therapeutic applications. It has a very high photocurrent and high sensitivity, meaning that substantial modulation can be achieved with light at intensities <10 mW cm$^{-2}$. The slow closing of this construct could enable tonic activation with pulsed light, further decreasing the optical dose into the tissue. The positive reversal potential (+16.6 mV) further contributes to the ability of this channelrhodopsin to depolarize cells, even when the cells are already partially depolarized. Together with its low proton permeability, these attributes make ChR2-3M a good candidate for therapeutic applications requiring slowly varying changes in optogenetic drive. However, the slow kinetics of ChR2-3M may limit its use for basic science applications such as circuit mapping. PsCatCh2.0 is more suitable for applications requiring precisely timed spikes.

## Materials and methods
### HEK293T cell culture
HEK293T cells were purchased from ATCC (CRL-3216) and validated by Short Tandem Repeat (STR) profiling. Mycoplasma testing was negative. Wild-type or engineered HEK293T cell lines were maintained at 37°C, 5% $CO_2$ in Dulbecco's modified Eagle medium supplemented with 10% fetal bovine serum, 1% GlutaMax-I, penicillin (100 U ml$^{-1}$), and streptomycin (100 µg ml$^{-1}$). For maintaining or expanding the cell culture, we used TC-treated culture dishes (Corning). For all imaging experiments, cells were plated on PDL-coated glass-bottomed dishes (Cellvis, Cat.# D35-14-1.5-N).

### Neuron culture
Primary E18 rat hippocampal neurons (fresh, never frozen, BrainBits #SDEHP) were dissociated following vendor protocols and plated in PDL-coated glass bottom dishes (Cellvis, Cat.# D35-14-1.5-N). Neurons (21 k cm$^{-2}$) were cocultured with primary rat glia (27 k cm$^{-2}$) to improve cell health and maturation.

### Lentivirus preparation
All the lentivirus preparations were made in house. HEK293T cells were co-transfected with the second-generation packaging plasmid psPAX2 (Addgene #12260), envelope plasmid VSV-G (Addgene #12259) and transfer plasmids at a ratio of 9:4:14. For small batches, 2.7 µg total plasmids for a small culture (300k cells in 35 mm dish) gave sufficient yield of lentivirus. Some viruses were concentrated using Lenti-X Concentrator (Takara Cat. # 631232) following vendor protocols and were concentrated 1/10. Quantities of virus used were quoted as non-concentrated amounts.

### Expression of optogenetic actuators and reporters
HEK293T cells were transduced at least 2 days before imaging with 50–200 µl of lentivirus encoding the desired channelrhodopsin. Cell lines were created for stable expression of pHoran4, and of pHoran4 with Dox-inducible $K_{ir}$2.1-CFP, using fluorescence-activated cell sorting on cells that already

had stable rtTA3 expression through antibiotic selection. $K_{ir}2.1$ expression was induced 2 days before imaging by adding 1 µg ml$^{-1}$ doxycycline, which was kept on the culture until time to image.

Neurons were transduced after 6–10 days in culture with (1) 200 µl lentivirus encoding pHoran4 driven by the cytomegalovirus (CMV) promoter, or 100–200 µl lentivirus encoding Quasar6a driven by the synapsin promoter and (2) 50–200 µl of the channelrhodopsin variants, also driven by the CMV promoter. Functional imaging was performed after 14–20 days in culture.

## pH calibration

The pH response of pHoran4 was calibrated by changing the buffer pH stepwise from 6.4 to 7.3 (*Figure 1—figure supplement 1*). To equilibrate the pH of the cytosol with the buffer pH, we added the K$^+$/H$^+$ exchanger nigericin at 14 µM. To prevent a [K$^+$] gradient from driving a proton gradient, we used a high-potassium extracellular buffer (*Thomas et al., 1979*; *Miesenböck et al., 1998*). The buffer composition was (in mM): Good's zwitterionic buffer 25, KCl 100, NaCl 38, CaCl$_2$ 1.8, MgSO$_4$ 0.8, NaH$_2$PO$_4$ 0.9. The Good buffer, chosen based on its pK$_a$ and effective buffering pH range, was 2-(*N*-morpholino)ethanesulfonic acid (MES) for pH 6.4 and (4-(2-hydroxyethyl)-1-piperazineethanesulfonic acid) (HEPES) for pH 6.7–7.3. After perfusion of the buffers with different pH values, we waited 1 min for the pH to equilibrate and recorded the steady-state fluorescence for each cell. $\Delta F/F$ was calculated using the pH 7.3 as the baseline. The $\Delta F/F$ at each pH was then averaged across cells, and this average was fit with piecewise linear interpolation, which was used for converting $\Delta F/F$ to pH in subsequent data analysis.

## Sample preparation for imaging

Before optical stimulation and imaging, 35 mm dishes were washed with 1 ml phosphate-buffered saline to remove residual culture medium, then filled with 2 ml extracellular (XC) buffer containing (in mM): 125 NaCl, 2.5 KCl, 2 CaCl$_2$, 1 MgCl$_2$, 15 HEPES, 25 glucose (pH 7.3). All imaging and electrophysiology were done using this XC buffer. For voltage imaging experiments in neurons, we added 10 µM NBQX, 20 µM Gabazine, 25 µM AP-V to block synaptic transmission.

BeRST1 was a gift from Evan Miller (Berkeley) and was used for voltage imaging in HEK cell monolayers. Cells were washed to remove culture medium and then incubated with 1–2 µM BeRST1 dye in XC buffer for 30 min. Immediately before imaging, samples were washed twice and immersed in XC buffer.

## Combined optogenetic stimulation and imaging

Experiments were conducted on a home-built inverted fluorescence microscope equipped with 405, 488, 532, 561, 594, and 640 nm laser lines and a scientific complementary metal–oxide semiconductor (CMOS) camera (Hamamatsu ORCA-Flash 4.0). Beams from lasers were combined using dichroic mirrors and sent through an acousto-optic tunable filter (Gooch and Housego TF525-250-6-3-GH18A) for temporal modulation of intensity of each wavelength. The beams were then expanded and sent either to a DMD (Vialux, V-7000 UV, 9515) for spatial modulation or sent directly into the microscope (to avoid power losses associated with the DMD). The beams were focused onto the back-focal plane of a ×60/1.2-NA (numerical aperture) water-immersion objective (Olympus UIS2 UPlanSApo ×60/1.20 W) or a ×20/0.75-NA objective (Olympus UIS2 UPlanSApo ×20/0.75). For Green and Yellow fluorescent protein, pHoran4, and QuasAr6a, fluorescence emission was separated from laser excitation using a dichroic mirror (488/561/633). Imaging of pHoran4 fluorescence was performed with 561 nm laser at illumination intensities of 100–200 mW cm$^{-2}$. Imaging of QuasAr6a fluorescence was performed with 640 nm laser at an illumination intensity of 8 W cm$^{-2}$. Stimulation of channelrhodopsins was performed with 488 nm laser at an illumination intensity of 400–800 mW cm$^{-2}$.

## Electrophysiology

For patch clamp measurements, filamented glass micropipettes (WPI) were pulled to a resistance of 5–10 MΩ and filled with internal solution containing (in mM): 6 NaCl, 130 K-aspartate, 2 MgCl$_2$, 5 CaCl$_2$, 11 ethylene glycol tetraacetic acid (EGTA), and 10 HEPES (pH 7.2). The patch electrode was controlled with a low-noise patch clamp amplifier (either A-M Systems model 2400 or Axon Instruments MultiClamp 700B). Current traces were collected in voltage clamp mode. The collected electrophysiology data had a moving average filter applied to help reduce noise. The time constants

were fit using single exponentials. In plots with multiple wavelengths of stimulation, the currents were normalized to the peak current for 488 nm stimulation.

### Wide-field imaging and patterning

Spatially resolved optical electrophysiology measurements were performed using a home-built upright ultra-wide-field microscope (*Werley et al., 2017*) with a large field of view ($4.6 \times 4.6$ mm$^2$, with $2.25 \times 2.25$ μm$^2$ pixel size in the sample plane) and high numerical aperture objective lens (Olympus MVPLAPO 2XC, NA 0.5). The fluorescence of BeRST1 was excited with a 639-nm laser (OptoEngine MLL-FN-639) at 100 mW cm$^{-2}$, illuminating the sample from below at an oblique angle to minimize background autofluorescence. BeRST1 fluorescence was separated from scattered laser excitation via a dichroic beam splitter (Semrock Di01-R405/488/561/635-t3–60x85) and an emission filter (Semrock FF01-708/75–60-D). Images were collected at a 100 Hz frame rate on a Hamamatsu Orca Flash 4.2 scientific CMOS camera. Optogenetic stimulation was performed by exciting Channelrhodopsins with a blue LED (Thorlabs M470L3) with a maximum intensity of 400 mW cm$^{-2}$.

### Measuring permeability of ChR2-3M

*Xenopus* oocytes were injected with cRNAs and maintained at 16°C for 2 days in ND96 solution: 96 mM NaCl, 2 mM KCl, 1 mM CaCl$_2$, 1 mM MgCl$_2$, 10 mM HEPES, pH 7.4, and 50 μg ml$^{-1}$ gentamycin. Two-electrode voltage clamp was used for oocyte electrophysiology with TURBO TEC-05 amplifier from NPI (NPI electronics GmbH, Tamm, Germany). For current amplitude comparisons, photocurrents were measured in extracellular solution containing 110 mM NaCl, 5 mM KCl, 2 mM MgCl$_2$, 2 mM BaCl$_2$, 5 mM HEPES, pH 7.6; holding at −70 mV. Shifts in reversal potential, $V_r$, were calculated by the reversal potential differences upon changing extracellular Na$^+$ or K$^+$ concentration from 120 mM (120 mM NaCl/KCl, 2 mM BaCl$_2$, 5 mM HEPES, pH adjusted to 7.6 by *N*-methyl-D-glucamine) to 1 mM (1 mM NaCl/KCl, 119 mM *N*-methyl-D-glucamine, 2 mM BaCl$_2$, 5 mM HEPES, pH adjusted to 7.6 by HCl). For all oocyte experiments, 473 nm laser at 5 mW mm$^{-2}$ was used for illumination.

### Data analysis

All data were processed and analyzed in MATLAB. For recordings with interleaved optogenetic stimulation and pH imaging, camera frames during stimulation were discarded, and frames during each period of imaging were averaged. Baseline fluorescence, $F_0$, was calculated from the first frame, before any optogenetic stimulation. A threshold on $F_0$ was set to restrict calculation of $\Delta F/F_0$ to signal-bearing regions of the sample.

Individual cells expressing the desired constructs were selected and fluorescence waveforms were calculated by averaging pixels whose baseline value exceeded the threshold. Sensor photoactivation artifacts were characterized using matched controls that expressed pHoran4 but no channelrhodopsin. Population-average photoartifacts were subtracted from the signals obtained from cells with channerhodopsin expression.

Dendrites were selected and analyzed in the same manner as somas, and were then associated with the connected soma. Acidification half-times were calculated by finding the maximum acidification during the stimulation period, and then finding the time point where the $\Delta F/F$ first reached half of the maximum decrease. Recovery after stimulation was fit to a single exponential, and the fit function was used to calculate the half-recovery time. Statistical tests were done using the Wilcoxon signed-rank test.

### Materials availability statement

Plasmids developed for this study are available on Addgene:
https://www.addgene.org/browse/article/28228806/

## Acknowledgements

We thank He Tian, Hillel Ori, Andrew Preecha, and Shahinoor Begum for helpful discussions and technical assistance. The BeRST1 dye was provided by Evan Miller. This work was supported by a Vannevar Bush Faculty Fellowship, and NIH grants 1-R01-MH117042 and 1-R01-NS126043.

## Additional information

### Funding

| Funder | Grant reference number | Author |
|---|---|---|
| Office of Naval Research | N00014-18-1-2859 | Adam E Cohen |
| National Institutes of Health | 1-R01-MH117042 | Rebecca Frank Hayward<br>F Phil Brooks III<br>Adam E Cohen |
| National Institutes of Health | 1-R01-NS126043 | Rebecca Frank Hayward<br>F Phil Brooks III<br>Adam E Cohen |

The funders had no role in study design, data collection, and interpretation, or the decision to submit the work for publication.

### Author contributions

Rebecca Frank Hayward, Conceptualization, Data curation, Investigation, Visualization, Methodology, Writing – original draft, Writing – review and editing; F Phil Brooks III, Investigation, Visualization; Shang Yang, Resources, Investigation; Shiqiang Gao, Resources, Supervision, Funding acquisition, Project administration; Adam E Cohen, Conceptualization, Data curation, Formal analysis, Supervision, Funding acquisition, Writing – original draft, Project administration, Writing – review and editing

### Author ORCIDs

Rebecca Frank Hayward ⓘ https://orcid.org/0009-0006-2132-7522
Shiqiang Gao ⓘ http://orcid.org/0000-0001-6190-9443
Adam E Cohen ⓘ https://orcid.org/0000-0002-8699-2404

Reviewer #1 (Public Review): https://doi.org/10.7554/eLife.86833.3.sa1
Reviewer #2 (Public Review): https://doi.org/10.7554/eLife.86833.3.sa2
Author Response https://doi.org/10.7554/eLife.86833.3.sa3

---

## Additional files

### Supplementary files

- MDAR checklist

### Data availability

Data underlying each figure panel are available on Figshare at: https://figshare.com/projects/Diminishing_neuronal_acidification_by_channelrhodopsins_with_low_proton_conduction/178173.

The following datasets were generated:

| Author(s) | Year | Dataset title | Dataset URL | Database and Identifier |
|---|---|---|---|---|
| Hayward RF | 2023 | Figure 4 Data | https://doi.org/10.6084/m9.figshare.24121323.v1 | figshare, 10.6084/m9.figshare.24121323.v1 |
| Hayward RF | 2023 | Figure 3 Data | https://doi.org/10.6084/m9.figshare.24121317.v1 | figshare, 10.6084/m9.figshare.24121317.v1 |
| Hayward RF | 2023 | Figure 2 Data | https://doi.org/10.6084/m9.figshare.24121305.v1 | figshare, 10.6084/m9.figshare.24121305.v1 |
| Hayward RF | 2023 | Figure 1 Data | https://doi.org/10.6084/m9.figshare.24121302.v1 | figshare, 10.6084/m9.figshare.24121302.v1 |

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
