## [Editor Report · eLife assessment]

This **important** and **compelling** study investigates the problem of intracellular acidification induced by commonly-used optogenetic stimulating opsins. The low proton permeability of two high-performance opsins is shown to reduce photostimulated acidification. The findings may be of broad interest in the fields of neuroscience research and optogenetic therapies.

---

## [Referee Report · Reviewer #1 (Public Review)]

In this manuscript, "Diminishing neuronal acidification by channelrhodopsins with low proton conduction" by Hayward and colleagues, the authors report on the properties of novel optogenetic tools, PsCatCh2.0 and ChR2-3M, that minimize photo-induced acidification. The authors point out that acidification is an undesirable side-effect of many optogenetic approaches that could be minimized using the new tools. ChRs are known to acidify cells, while Arch are known to alkalize cells. This becomes particularly important when optical stimulation is prolonged and pH changes can become significant. pH is known to affect neuronal excitability, vesicular release, and more. To develop novel optogenetic tools with minimal proton conductances, the authors combined channelrhodopsin stimulation with a red-shifted pH sensor to measure pH during optogenetic stimulation. The authors report that optogenetic activation of CheRiff caused slow cellular acidification. 150 seconds of illumination caused a 3-fold increase in protons or approximately a 0.6 unit pH change that returned to baseline very slowly. They also found that pH changes occurred more rapidly, and recovered more rapidly, in dendrites. The authors go on to robustly characterize PsCatCh2.0 and ChR2-3M in terms of their proton conductances, photocurrent, kinetics, and more. They convincingly show that these constructs induced reduced acidification while maintaining robust photocurrents. In sum, this manuscript shows important findings that convincingly characterizes 2 optogenetic tools that have reduced pH artifacts that may be of broad interest to the field of neuroscience research and optogenetic therapies.

---

## [Referee Report · Reviewer #2 (Public Review)]

In this paper, the authors utilize optogenetic stimulation and imaging techniques with fluorescent reporters for pH and membrane voltage to examine the extent of intracellular acidification produced by different ion-conducting opsins. The commonly used opsin CheRiff is found to conduct enough protons to alter intracellular pH in soma and dendrites of targeted neurons and in monolayers of HEK293T cells, whereas opsins ChR2-3M and PsCatCh2.0 are shown to produce negligible changes in intracellular pH as their photocurrents are mostly carried by metal cations. The conclusion that ChR2-3M and PsCatCh2.0 are more suited than proton conducting opsins for optogenetic applications is well supported by the data.

---

## [Author Response]

The following is the authors’ response to the original reviews.

We thank the reviewers for these helpful and thoughtful comments.

**Reviewer #1 (Recommendations For The Authors):**
Major comments:What was the nature of the 0.1 increase in pH caused by illumination in CheRiff-negative cells? Is this thought to be a temperature effect?

The increase in pHoran4 fluorescence in CheRiff-negative cells is most likely not from a pH change; rather, it most likely reflects blue light-mediated photoactivation of the mOrange-derived chromophore in pHoran4. Similar photoartifacts have been reported in other fluorescent protein reporters (see e.g. Farhi, Samouil L., et al. "Wide-area all-optical neurophysiology in acute brain slices." Journal of Neuroscience 39.25 (2019): 4889-4908.).

The baseline measurement in CheRiff-negative cells is to control for this type of artifact. We subtract the mean signal from the CheRiff-negative cells to correct the signals from the CheRiff-positive cells, as described in the Main Text.

Does Kir2.1 have a proton conductance? Was the resting pH of HEK cells changed by Kir2.1 expression? Fig 2D suggest basal pH is equivalent +/- Kir2.1 but it would be good to show that data.

This is an interesting question which our data do not answer conclusively. Since we used an intensiometric (as opposed to ratiometric) pH indicator, our measurements only provide relative pH changes. We assumed a constant initial pH. We have revised the text to make clear that this is an assumption.

Prior studies of pH-dependent Kir2.1 activity did not find evidence of a proton current (i.e. no change in current upon extracellular acidification), though the channel is closed by intracellular acidification. See: Ye, Wenlei, et al. "The K+ channel KIR2. 1 functions in tandem with proton influx to mediate sour taste transduction." Proceedings of the National Academy of Sciences 113.2 (2016): E229-E238. We added this information to the text.

The pKa of pHoran4 is 7.5, so a decrease in initial pH would decrease the slope of F vs pH. We observed higher (absolute value) ΔF/F in the Kir2.1 expressing cells than in the non-expressing cells, confirming that the Kir2.1-expressing cells had larger CheRiff-mediated acidification than the Kir2.1-negative cells (Figure 2D). Thus this conclusion remains true regardless of whether Kir2.1 has a proton conductance.

What channels/transporter mediate proton flux in CheRiff + Kir2.1 experiments? Is the increased proton flux simply due to more H+ ions passing through CheRiff when cells are hyperpolarized or may other voltage-dependent processes effect pH?

Fig. 2G-M address this question, specifically. We targeted the blue light in a “zebra” pattern to only activate CheRiff in a subset of cells. We then used voltage imaging to show that the induced voltage spread over a much wider area than the blue-illuminated region, due to gap junction coupling between the cells. If protons flowed through some voltage-dependent channel other than CheRiff, then we would expect the acidification to follow the voltage profile. If protons primarily flowed through the CheRiff, then we would expect the acidification to follow the illumination profile. Fig. 2K and the following quantification show clearly that the acidification followed the illumination profile, and hence the proton current was primarily through CheRiff.

Is Kir2.1 included in the spatial illumination experiments (Fig. 2G-M)? If so, it would be helpful to note it. The color scheme suggest it is but it would be good to note it explicitly.

Yes. Clarified in text.

Why is the acidification caused by 10 second of illumination smaller in Fig 2L, as compared to the equivalent experiment in 2D? Is this due to the spatial nature of the illumination? It seems that the pH change at the site of illumination should be equivalent between these 2 experiments.

The illumination protocol between the two experiments has different duty cycles (compare Fig. 2C and 2J), so the time-averaged intensity is different. There can also be batch-to-batch variation in CheRiff expression which would alter the proton flux and thus pH change. To control for this, comparisons were always made between batches of cells prepared together.

The authors used 150 second illumination to examine pH changes but only 13.5 seconds to differentiate between pH changes caused by the light-activated conductance and those secondary to depolarization. Would pH changes lose their spatial limitations if a similar 150 second illumination was used? This is important because the pH change seen in the "Blue On" region was quite small.

Yes, protons can diffuse between cells via gap junctions, smoothing out the spatial structure of the pH over long times. See e.g. Wu, Ling, et al. "PARIS, an optogenetic method for functionally mapping gap junctions." Elife 8 (2019): e43366.

We used a short (13.5 s) protocol specifically to distinguish CheRiff-mediated acidification from acidification via other conductances in electrically coupled neighboring cells. If we had waited for longer, lateral proton diffusion could have muddied the interpretation of these experiments.

How long do action potentials shown in between illuminations in Fig 4H (ChR2 3M) last following cessation of illumination?

The closing time, τoff, of the Channelrhodopsins are shown in Table 1. The ChR2-3M has an off-time of almost 2 seconds. The duration of post-stimulus persistent firing is expected to depend on the expression level of the ChR2-3M, the strength of the optogenetic stimulus and the excitation threshold of the neurons, i.e. on how far above threshold the neuron is at the moment the blue light turns off. Thus we expect the post-stimulus firing time to be highly variable between cells and also to depend on optogenetic stimulus strength. In our experiments action potentials were observed throughout the 0.5 s dark interval between stimuli.

While ChR2-3M construct may have promise for therapeutic applications, those strengths limit its use or basic science applications like circuit mapping. This should be noted in the discussion.

Ok. We now mention this in the discussion.

Please define EPD50 within the text of the results section.

Ok. Fixed.

**Reviewer #2 (Recommendations For The Authors):**
This is an interesting manuscript investigating a potential limitation of optogenetic manipulation of cell excitability and its solution. The work is conducted rigorously and explained clearly. I only have minor concerns:I think the impact of the study could be broadened by examining additional proton permeable opsins for their effects on intracellular pH. A single assay could be used to compare different opsins to CheRiff and show that the problem of intracellular acidification is not limited to CheRiff.

Yes, this is interesting. There are so many opsins and illumination protocols in use that we could not do an exhaustive characterization; we encourage people to test their own opsin under their conditions if doing chronic simulation. The plasmid constructs used for this work are available on Addgene.

I am not clear on what Figure S3A is showing because I cannot see a patterning like the one shown in Fig. 2H. Perhaps a higher magnification could solve the problem.

Figure S3A does not have the zebra-striped pattern of Figure 2H. In Fig S3A, we used just one column of illumination. The point was to test the ability of each opsin to depolarize the HEK cells. We added images of the illumination pattern and adjusted the caption to make this clear.

When discussing the sustained photocurrent of PsCatCh2.0, a reference to Govorunova et al. J. Biol. Chem. 2013 should be added as the low extent of light induced inactivation appears to be, at least in part, a characteristic of the particular type of opsin from P. subcordiformis.

Added reference.